

# Temporal dynamics of requirements engineering from mobile app reviews

Vitor Mesaque Alves de Lima[1], Adailton Ferreira de Araújo[2] and Ricardo Marcondes Marcacini[1,2]

[1] Faculty of Computing (FACOM), Federal University of Mato Grosso do Sul (UFMS), Campo Grande, Mato Grosso do Sul, Brazil
[2] Institute of Mathematics and Computer Sciences (ICMC), University of São Paulo (USP), São Carlos, São Paulo, Brazil

## ABSTRACT

Opinion mining for app reviews aims to analyze people's comments from app stores to support data-driven requirements engineering activities, such as bug report classification, new feature requests, and usage experience. However, due to a large amount of textual data, manually analyzing these comments is challenging, and machine-learning-based methods have been used to automate opinion mining. Although recent methods have obtained promising results for extracting and categorizing requirements from users' opinions, the main focus of existing studies is to help software engineers to explore historical user behavior regarding software requirements. Thus, existing models are used to support corrective maintenance from app reviews, while we argue that this valuable user knowledge can be used for preventive software maintenance. This paper introduces the temporal dynamics of requirements analysis to answer the following question: how to predict initial trends on defective requirements from users' opinions before negatively impacting the overall app's evaluation? We present the MAPP-Reviews (Monitoring App Reviews) method, which (i) extracts requirements with negative evaluation from app reviews, (ii) generates time series based on the frequency of negative evaluation, and (iii) trains predictive models to identify requirements with higher trends of negative evaluation. The experimental results from approximately 85,000 reviews show that opinions extracted from user reviews provide information about the future behavior of an app requirement, thereby allowing software engineers to anticipate the identification of requirements that may affect the future app's ratings.

## INTRODUCTION

Opinions extracted from app reviews provide a wide range of user feedback to support requirements engineering activities, such as bug report classification, new feature requests, and usage experience (*Dabrowski et al., 2020*; *Martin et al., 2016*; *AlSubaihin et al., 2019*; *Araujo & Marcacini, 2021*). However, manually analyzing a reviews dataset to extract useful knowledge from the opinions is challenging because of the large amount of data and the high frequency of new reviews published by users (*Johanssen et al., 2019*; *Martin et al., 2016*). To deal with these challenges, opinion mining has been increasingly used for

Corresponding author
Vitor Mesaque Alves de Lima,
vitor.lima@ufms.br

computational analysis of the people's opinions from free texts (*Liu, 2012*). In the context of app reviews, opinion mining allows extracting excerpts from comments and mapping them to software requirements, as well as classifying the positive, negative or neutral polarity of these requirements according to the users' experience (*Dabrowski et al., 2020*).

One of the main challenges for software quality maintenance is identifying emerging issues, *e.g.*, bugs, in a timely manner (*April & Abran, 2012*). These issues can generate huge losses, as users can fail to perform important tasks or generate dissatisfaction that leads the user to uninstall the app. A recent survey showed that 78.3% of developers consider removing unnecessary and defective requirements to be equally or more important than adding new requirements (*Nayebi et al., 2018*). According to *Lientz & Swanson (1980)*, maintenance activities are categorized into four classes: (i) adaptive – changes in the software environment; (ii) perfective – new user requirements; (iii) corrective – fixing errors; and (iv) preventive – prevent problems in the future. The authors showed that around 21% of the maintenance effort was on the last two types (*Bennett & Rajlich, 2000*). Specifically, in the context of mobile apps *Mcilroy, Ali & Hassan (2016)* found that rationale for the update most frequently communicated task in app stores is bug fixing which occurs in 63% of the updates. Thus, approaches that automate the analysis of potentially defective software requirements from app reviews are important to make strategic updates, as well as prioritization and planning of new releases (*Licorish, Savarimuthu & Keertipati, 2017*). In addition, the app stores offer a more dynamic way of distributing the software directly to users, with shorter release times than traditional software systems, *i.e.*, continuous update releases are performed every few weeks or even days (*Nayebi, Adams & Ruhe, 2016*). Therefore, app reviews provide quick feedback from the crowd about software misbehavior that may not necessarily be reproducible during regular development/testing activities, *e.g.*, device combinations, screen sizes, operating systems and network conditions (*Palomba et al., 2018*). This continuous crowd feedback can be used by developers in the development and preventive maintenance process.

Using an opinion mining approach, we argue that software engineers can investigate bugs and misbehavior early when an app receives negative reviews. Opinion mining techniques can organize reviews based on the identified software requirements and their associated user's sentiment (*Dabrowski et al., 2020*). Consequently, developers can examine negative reviews about a specific feature to understand the user's concerns about a defective requirement and potentially fix it more quickly, *i.e.*, before impacting many users and negatively affecting the app's ratings.

Different strategies have recently been proposed to discover these emerging issues (*Zhao et al., 2020*), such as issues categorization (*Tudor & Walter, 2006*; *Iacob & Harrison, 2013*; *Galvis Carreño & Winbladh, 2013*; *Pagano & Maalej, 2013*; *Mcilroy et al., 2016*, *Khalid et al., 2015*; *Panichella et al., 2015*; *Panichella et al., 2016*), sentiment analysis of the software requirements to identify certain levels of dissatisfaction (*Gao et al., 2020*), and analyze the degree of utility of a requirement (*Guzman & Maalej, 2014*). These approaches are concerned only with past reviews and acting in a corrective way, *i.e.*, these approaches do not have preventive strategies to anticipate problems that can become frequent and impact more users in the coming days or weeks. Analyzing the temporal dynamics of a

requirement from app reviews provides information about a requirement's future behavior. In this sense, we raise the following research question: how do we predict initial trends on defective requirements from users' opinions before negatively impacting the overall app's evaluation?

In this paper, we present the MAPP-Reviews (Monitoring App Reviews) method. MAPP-Reviews explores the temporal dynamics of software requirements extracted from app reviews. First, we collect, pre-process and extract software requirements from large review datasets. Then, the software requirements associated with negative reviews are organized into groups according to their content similarity by using clustering technique. The temporal dynamics of each requirement group is modeled using a time series, which indicates the time frequency of a software requirement from negative reviews. Finally, we train predictive models on historical time series to forecast future points. Forecasting is interpreted as signals to identify which requirements may negatively impact the app in the future, *e.g.*, identify signs of app misbehavior before impacting many users and prevent the low app ratings. Our main contributions are briefly summarized below:

- Although there are promising methods for extracting candidate software requirements from application reviews, such methods do not consider that users describe the same software requirement in different ways with non-technical and informal language. Our MAPP-Reviews method introduces software requirements clustering to standardize different software requirement writing variations. In this case, we explore contextual word embeddings for software requirements representation, which have recently been proposed to support natural language processing. When considering the clustering structure, we can more accurately quantify the number of negative user mentions of a software requirement over time.

- We present a method to generate the temporal dynamics of negative ratings of a software requirements cluster by using time series. Our method uses equal-interval segmentation to calculate the frequency of software requirements mentions in each time interval. Thus, a time series is obtained and used to analyze and visualize the temporal dynamics of the cluster, where we are especially interested in intervals where sudden changes happen.

- Time series forecasting is useful to identify in advance an upward trend of negative reviews for a given software requirement. However, most existing forecasting models do not consider domain-specific information that affects user behavior, such as holidays, new app releases and updates, marketing campaigns, and other external events. In the MAPP-Reviews method, we investigate the incorporation of software domain-specific information through trend changepoints. We explore both automatic and manual changepoint estimation.

We carried out an experimental evaluation involving approximately 85,000 reviews over 2.5 years for three food delivery apps. The experimental results show that it is possible to find significant points in the time series that can provide information about the future behavior of the requirement through app reviews. Our method can provide important

information to software engineers regarding software development and maintenance. Moreover, software engineers can act preventively through the proposed MAPP-Reviews approach and reduce the impacts of a defective requirement.

This paper is structured as follows. Section "Background and Related Work" presents the literature review and related work about mining user opinions to support requirement engineering and emerging issue detection. In "MAPP-Reviews method" section, we present the architecture of the proposed method. We present the main results in "Results" section. Thereafter, we evaluate and discuss the main findings of the research in "Discussion" section. Finally, in "Conclusions" section, we present the final considerations and future work.

## BACKGROUND AND RELATED WORK

The opinion mining of app reviews can involve several steps, such as software requirements organization from reviews (*Araujo & Marcacini, 2021*), grouping similar apps using textual features (*Al-Subaihin et al., 2016*; *Harman, Jia & Zhang, 2012*), reviews classification in categories of interest to developers (*e.g.*, Bug and New Features) (*Araujo et al., 2020*), sentiment analysis of the users' opinion about the requirements (*Dragoni, Federici & Rexha, 2019*; *Malik, Shakshuki & Yoo, 2020*), and the prediction of the review utility score (*Zhang & Lin, 2018*). The requirements extraction has an essential role in these steps since the failure in this task directly affects the performance of the other steps.

*Dabrowski et al. (2020)* evaluated the performance of the three state-of-the-art requirements extraction approaches: SAFE (*Johann et al., 2017*), ReUS (*Dragoni, Federici & Rexha, 2019*) and GuMa (*Guzman & Maalej, 2014*). These approaches explore rule-based information extraction from linguistic features. GuMa (*Guzman & Maalej, 2014*) used a co-location algorithm, thereby identifying expressions of two or more words that correspond to a conventional way of referring to things. SAFE (*Johann et al., 2017*) and ReUS (*Dragoni, Federici & Rexha, 2019*) defined linguistic rules based on grammatical classes and semantic dependence. The experimental evaluation of *Dabrowski et al. (2020)* revealed that the low accuracy presented by the rule-based approaches could hinder its use in practice.

*Araujo & Marcacini (2021)* proposed RE-BERT (Requirements Engineering using Bidirectional Encoder Representations from Transformers) method for software requirements extraction from reviews based on Local Context Word Embeddings (*i.e.* deep neural language model). RE-BERT models the requirements extraction as a token classification task from deep neural networks. To solve some limitations of rule-based approaches, RE-BERT allows the generation of word embeddings for reviews according to the context of the sentence in which the software requirement occurs. Moreover, RE-BERT explores a multi-domain training strategy to enable software requirements extraction from app reviews of new domains without labeled data.

After extracting requirements from app reviews, there is a step to identify more relevant requirements and organize them into groups of similar requirements. Traditionally, requirements obtained from user interviews are prioritized with manual analysis techniques, such as the MoSCoW (*Tudor & Walter, 2006*) method that categorizes each

requirement into groups, and applies the AHP (Analytical Hierarchy Process) decision-making (*Saaty, 1980*). These techniques are not suitable for prioritizing large numbers of software requirements because they require domain experts to categorize each requirement. Therefore, recent studies have applied data mining approaches and statistical techniques (*Pagano & Maalej, 2013*).

The statistical techniques have been used to find issues such as to examine how app features predict an app's popularity (*Chen & Liu, 2011*), to analyze the correlations between the textual size of the reviews and users' dissatisfaction (*Vasa et al., 2012*), lower rating and negative sentiments (*Hoon et al., 2012*), correlations between the rating assigned by users and the number of app downloads (*Harman, Jia & Zhang, 2012*), to the word usage patterns in reviews (*Gómez et al., 2015*; *Licorish, Savarimuthu & Keertipati, 2017*), to detect traceability links between app reviews and code changes addressing them (*Palomba et al., 2018*), and explore the feature lifecycles in app stores (*Sarro et al., 2015*). There also exists some work focus on defining taxonomies of reviews to assist mobile app developers with planning maintenance and evolution activities (*Di Sorbo et al., 2016*; *Ciurumelea et al., 2017*; *Nayebi et al., 2018*). In addition to user reviews, previous works (*Guzman, Alkadhi & Seyff, 2016*, *2017*; *Nayebi, Cho & Ruhe, 2018*) explored how a dataset of tweets can provide complementary information to support mobile app development.

From a labeling perspective, previous works classified and grouped software reviews into classes and categories (*Iacob & Harrison, 2013*; *Galvis Carreño & Winbladh, 2013*; *Pagano & Maalej, 2013*; *Mcilroy et al., 2016*, *Khalid et al., 2015*; *Chen et al., 2014*; *Gómez et al., 2015*; *Gu & Kim, 2015*; *Maalej & Nabil, 2015*; *Villarroel et al., 2016*; *Nayebi et al., 2017*), such as feature requests, requests for improvements, requests for bug fixes, and usage experience. *Noei, Zhang & Zou (2021)* used topic modeling to determine the key topics of user reviews for different app categories.

Regarding analyzing emerging issues from app reviews, existing studies are usually based on topic modeling or clustering techniques. For example, LDA (Latent Dirichlet Allocation) (*Blei, Ng & Jordan, 2003*), DIVER (iDentifying emerging app Issues *Via* usER feedback) (*Gao et al., 2019*) and IDEA (*Gao et al., 2018*) approaches were used for app reviews. The LDA approach is a topic modeling method used to determine patterns of textual topics, *i.e.*, to capture the pattern in a document that produces a topic. LDA is a probabilistic distribution algorithm for assigning topics to documents. A topic is a probabilistic distribution over words, and each document represents a mixture of latent topics (*Guzman & Maalej, 2014*). In the context of mining user opinions in app reviews, especially to detect emerging issues, the documents in the LDA are app reviews, and the extracted topics are used to detect emerging issues. The IDEA approach improves LDA by considering topic distributions in a context window when detecting emerging topics by tracking topic variations over versions (*Gao et al., 2020*). In addition, the IDEA approach implements an automatic topic interpretation method to label each topic with the most representative sentences and phrases (*Gao et al., 2020*). In the same direction, the DIVER approach was proposed to detect emerging app issues, but mainly in beta test periods (*Gao et al., 2019*). The IDEA, DIVER and LDA approaches have not been considered sentiment of user reviews. Recently, the MERIT (iMproved EmeRging Issue deTection)

(*Gao et al., 2020*) approach was proposed and explore word embedding techniques to prioritize phrases/sentences of each positive and negative topic. *Phong et al. (2015)* and *Vu et al. (2016)* grouped the keywords and phrases using clustering algorithms and then determine and monitor over time the emergent clusters based on the occurrence frequencies of the keywords and phrases in each cluster. *Palomba et al. (2015)* proposes an approach to tracking informative user reviews of source code changes and to monitor the extent to which developers addressing user reviews. These approaches are descriptive models, *i.e.*, they analyze historical data to interpret and understand the behavior of past reviews. In our paper, we are interested in predictive models that aim to anticipate the growth of negative reviews that can impact the app's evaluation.

In short, app reviews formed the basis for many studies and decisions ranging from feature extraction to release planning of mobile apps. However, previous related works do not explore the temporal dynamics with a predictive model of requirements in reviews, as shown in Table 1. Related works that incorporate temporal dynamics cover only descriptive models. In addition, existing studies focus on only a few steps of the opinion mining process from app reviews, which hinders its use in real-world applications. Our proposal instantiates a complete opinion mining process and incorporates temporal dynamics of software requirements extracted from app reviews into forecasting models to address these drawbacks.

## THE MAPP-REVIEWS METHOD

In order to analyze the temporal dynamics of software requirements, we present the MAPP-Reviews approach with five stages, as shown in Fig. 1. First, we collect mobile app reviews in app stores through a web crawler. Second, we group the similar extracted requirements by using clustering methods. Third, the most relevant clusters are identified to generate time series from negative reviews. Finally, we train the predictive model from time series to forecast software requirements involved with negative reviews, which will potentially impact the app's rating.

### App reviews

The app stores provide the textual content of the reviews, the publication date, and the rating stars of user-reported reviews. In the first stage of MAPP-Reviews, raw reviews are collected from the app stores using a web crawler tool through a RESTful API. At this stage, there is no pre-processing in the textual content of reviews. Data is organized in the appropriate data structure and automatically batched to be processed by the requirements extraction stage of MAPP-Reviews. In the experimental evaluation presented in this article, we used reviews collected from three food delivery apps: Uber Eats, Foodpanda, and Zomato.

### Requirements extraction

This section describes stages 2 of the MAPP-Reviews method, where there is the software requirements extraction from app reviews and text pre-processing using contextual word embeddings.

**Table 1 Overview of related works.**

| References | Data representation | Pre-processing and extraction of requirements | Requirements/topics clustering and labeling | Temporal dynamics |
|---|---|---|---|---|
| *Araujo & Marcacini (2021)* | Word embeddings | Token classification | No. | No. |
| *Gao et al. (2020)* | Word embeddings | Rule-based and topic modeling | Yes. It combines word embeddings with topic distributions as the semantic representations of words | Yes. Descriptive model |
| *Malik, Shakshuki & Yoo (2020)* | Bag-of-words | Rule-based | No. | No. |
| *Gao et al. (2019)* | Vector space | Rule-based and topic modelling | Yes. Anomaly clustering algorithm | Yes. Descriptive model |
| *Dragoni, Federici & Rexha (2019)* | Dependency tree | Rule-based | No. | No. |
| *Gao et al. (2018)* | Probability vector | Rule-based and topic modelling | Yes. AOLDA – Adaptively Online Latent Dirichlet allocation. The topic labeling method considers the semantic similarity between the candidates and the topics | Yes. Descriptive model |
| *Johann et al. (2017)* | Keywords | Rule-based | No. | No. |
| *Vu et al. (2016)* | Word embeddings | Pre-defined | Yes. Soft clustering algorithm that uses vector representation of words from Word2vec | Yes. Descriptive model |
| *Villarroel et al. (2016)* | Bag-of-words | Rule-based | Yes. DBSCAN clustering algorithm. Each cluster has a label composed of the five most frequent terms | No. |
| *Gu & Kim (2015)* | Semantic dependence graph | Rule-based | Yes. Clustering aspect-opinion pairs with the same aspects | Yes. Descriptive model |
| *Phong et al. (2015)* | Vector space | Rule-based | Yes. Word2vec and K-means | Yes. Descriptive model |
| *Guzman & Maalej (2014)* | Keywords | Rule-based and topic modeling | Yes. LDA approach | No. |
| *Chen et al. (2014)* | Bag-of-words | Topic modelling | Yes. LDA and ASUM approach with labelling | Yes. Descriptive model |
| *Iacob & Harrison (2013)* | Keywords | Rule-based and topic modelling | Yes. LDA approach | No. |
| *Galvis Carreño & Winbladh (2013)* | Bag-of-words | Topic modelling | Yes. Aspect and Sentiment Unification Model (ASUM) approach | No. |
| *Harman, Jia & Zhang (2012)* | Keywords | Pre-defined | Yes. Greedy-based clustering algorithm | No. |
| *Palomba et al. (2018)* | Bag-of-words | Topic-modeling | Yes. AR-Miner approach with labeling | No. |

MAPP-Reviews uses the pre-trained RE-BERT (*Araujo & Marcacini, 2021*) model to extract software requirements from app reviews. RE-BERT is an extractor developed from our previous research. We trained the RE-BERT model using a labeled reviews dataset generated with a manual annotation process, as described by *Dabrowski et al. (2020)*. The reviews are from 8 apps of different categories as showed in Table 2. RE-BERT uses a cross-domain training strategy, where the model was trained in 7 apps and tested in one unknown app for the test step. RE-BERT software requirements extraction performance was compared to SAFE (*Johann et al., 2017*), ReUS (*Dragoni, Federici & Rexha, 2019*) and GuMa (*Guzman & Maalej, 2014*). Since RE-BERT uses pre-trained models for semantic

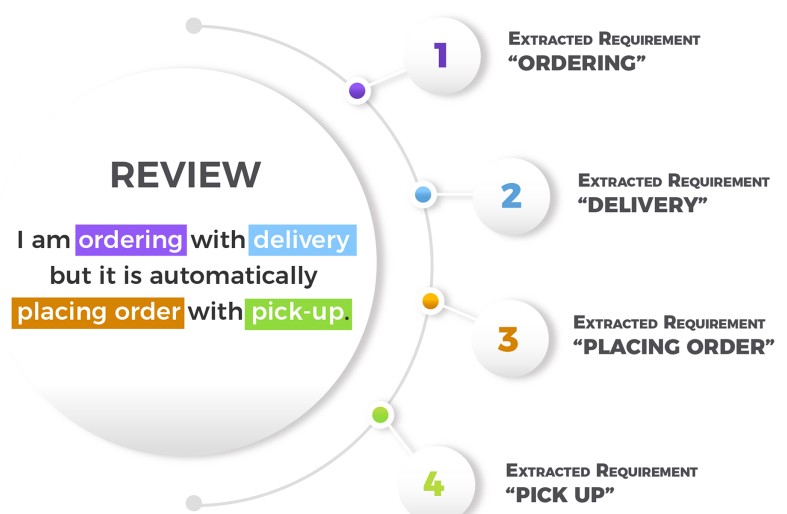

**MAPP-Reviews**

**STAGE 1**
App
Reviews

Raw reviews are collected from
the app stores.

**STAGE 2**
Requirements
Extraction

RE-BERT model to extract
software requirements from
app reviews..

**STAGE 3**
Requirements
Clustering

K-means algorithm to obtain a
clustering model of semantically
similar software requirements.

**STAGE 4**
Time Series
Generation

Time series for each software
requirements cluster.

**STAGE 5**
Predictive
Models

Prophet forecasting model to
identify requirements with
higher trends of negative
evaluation.

**Figure 1 Overview of the proposed method for analyzing temporal dynamics of requirements engineering from mobile app reviews.**

**Table 2 Statistics about the datasets from 8 apps of different categories used to train the RE-BERT model.**

|  | eBay | Evernote | Facebook | Netflix | Photo editor | Spotify | Twitter | WhatsApp |
|---|---|---|---|---|---|---|---|---|
| Reviews | 1,962 | 4,832 | 8,293 | 14,310 | 7,690 | 14,487 | 63,628 | 248,641 |
| Category | Shopping | Productivity | Social | Entertainment | Photography | Music and Audio | Social | Communication |

**REVIEW**

I am ordering with delivery
but it is automatically
placing order with pick-up.

1 EXTRACTED REQUIREMENT
"ORDERING"

2 EXTRACTED REQUIREMENT
"DELIVERY"

3 EXTRACTED REQUIREMENT
"PLACING ORDER"

4 EXTRACTED REQUIREMENT
"PICK UP"

**Figure 2 Example of a review and extracted requirements.**

representation of texts, the extraction performance is significantly superior to the rule-based methods. Given this scenario, we selected RE-BERT for the requirement extraction stage. Figure 2 shows an example of review and extracted software requirements. In the raw review *"I am ordering with delivery but it is automatically placing order with pick-up"*,

four software requirements were extracted ("ordering", "delivery", "placing order", and "pick-up"). Note that "placing order" and "ordering" are the same requirement in practice. In the clustering step of the MAPP-Reviews method, these requirements are grouped in the same cluster, as they refer to the same feature.

RE-BERT returns the probability that each token (*e.g.* word) is a software requirement. Consecutive tokens in a sentence are concatenated to obtain software requirements expressions composed of two or more tokens. We filter reviews that are more associated with negative comments through user feedback. Consider that the user gives a star rating when submitting a review for an app. Generally, the star rating ranges from 1 to 5. This rating can be considered as the level of user satisfaction. In particular, we are interested in defective software requirements, and only reviews with 1 or 2 rating stars were considered. Thus, we use RE-BERT to extract only software requirements mentioned in reviews that may involve complaints, bad usage experience, or malfunction of app features.

RE-BERT extracts software requirements directly from the document reviews and we have to deal with the drawback that the same requirement can be written in different ways by users. Thus, we propose a software requirement semantic clustering, in which different writing variations of the same requirement must be standardized. However, the clustering step requires that the texts be pre-processed and structured in a format that allows the calculation of similarity measures between requirements.

We represent each software requirement through contextual word embedding. Word embeddings are vector representations for textual data in an embedding space, where we can compare two texts semantically using similarity measures. Different models of word embeddings have been proposed, such as Word2vec (*Mikolov et al., 2013*), Glove (*Pennington, Socher & Manning, 2014*), FastText (*Bojanowski et al., 2017*) and BERT (*Devlin et al., 2018*). We use the BERT Sentence-Transformers model (*Reimers & Gurevych, 2019*) to maintain an neural network architecture similar to RE-BERT. BERT is a contextual neural language model, where for a given sequence of tokens, we can learn a word embedding representation for a token. Word embeddings can calculate the semantic proximity between tokens and entire sentences, and the embeddings can be used as input to train the classifier. BERT-based models are promising to learn contextual word embeddings from long-term dependencies between tokens in sentences and sentences (*Araujo & Marcacini, 2021*). However, we highlight that a local context more impacts the extraction of software requirements from reviews, *i.e.*, tokens closer to those of software requirements are more significant (*Araujo & Marcacini, 2021*). Therefore, RE-BERT explores local contexts to identify relevant candidates for software requirements. Formally, let $E = \{r_1, r_2, \ldots, r_n\}$ be a set of $n$ extracted software requirements, where $r_i = (t_1, \ldots, t_k)$ are a sequence of $k$ tokens of the requirement $r_i$. BERT explore a masked language modeling procedure, *i.e.*, BERT model first generates a corrupted $\hat{x}$ version of the sequence, where approximately 15% of the words are randomly selected to be replaced by a special token called (MASK) (*Araujo & Marcacini, 2021*). One of the training objectives is the noisy reconstruction defined in Eq. (1),

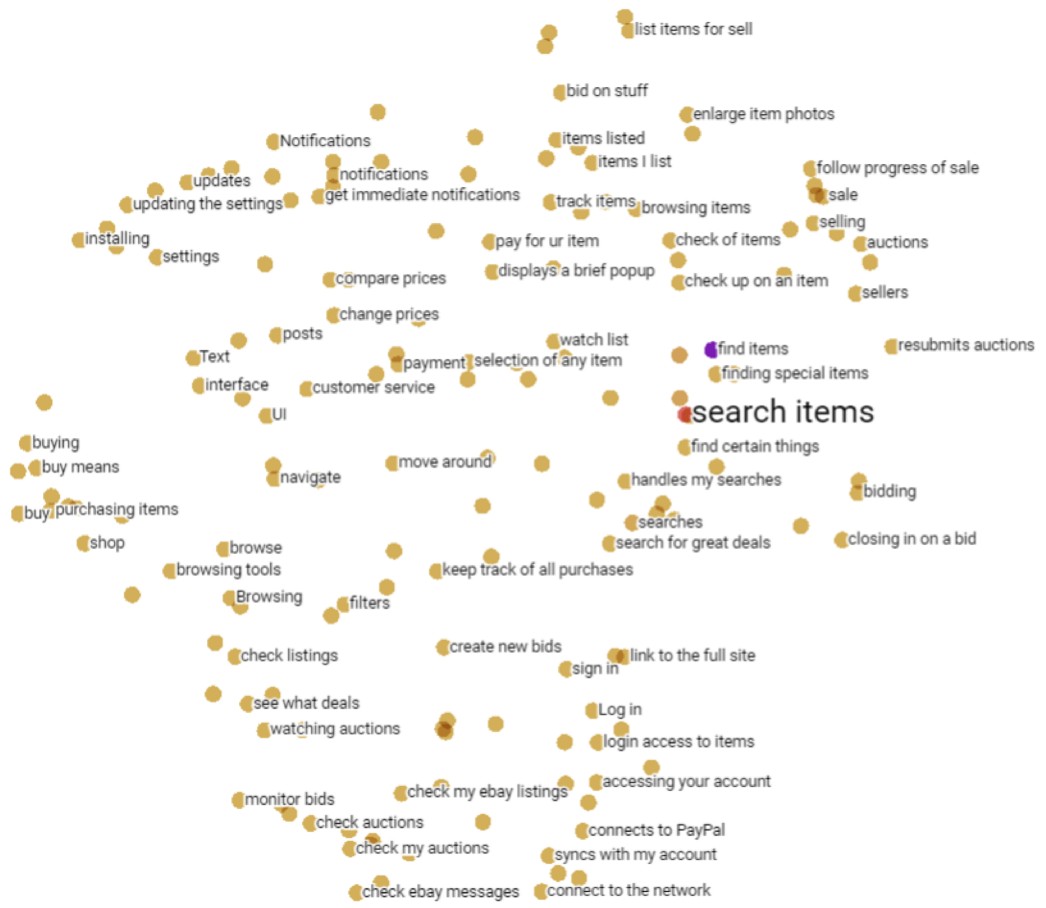

**Figure 3 Set of software requirements in a two-dimensional space obtained from contextual word embeddings.**               

$$p(\bar{r}|\hat{r}) = \sum_{j=1}^{k} m_j \frac{exp(\mathbf{h}_{c_j}^{T}\mathbf{w}_{t_j})}{\sum_{t'} exp(\mathbf{h}_{c_j}^{T}\mathbf{w}_{t'})} \tag{1}$$

where $\hat{r}$ is a corrupted token sequence of requirement $r$, $\bar{r}$ is the masked tokens, $m_t$ is equal to 1 when $t_j$ is masked and 0 otherwise. The $c_t$ represents context information for the token $t_j$, usually the neighboring tokens. We extract token embeddings from the pre-trained BERT model, where $\mathbf{h}_{c_j}$ is a context embedding and $\mathbf{w}_{t_j}$ is a word embedding of the token $t_j$. The term $\sum_{t'} exp(\mathbf{h}_c^{T}\mathbf{w}_{t'})$ is a normalization factor using all tokens $t'$ from a context $c$. BERT uses the Transformer deep neural network to solve $p(\bar{r}|\hat{r})$ of the Eq. (1). Figure 3 illustrates a set of software requirements in a two-dimensional space obtained from contextual word embeddings. Note that the vector space of embeddings preserves the proximity of similar requirements, but written in different ways by users such as "*search items*", "*find items*", "*handles my searches*" and "*find special items*".

**Table 3 Example of software requirement cluster "Payment" and some tokens allocated in the cluster with their respective silhouette values.**

| Cluster label | Tokens with silhouette ($s$) |
|---|---|
| Payment | "payment getting" ($s$ = 0.2618), "payment get" ($s$ = 0.2547), "getting payment" ($s$ = 0.2530), "take payment" ($s$ = 0.2504), "payment taking" ($s$ = 0.2471), "payment" ($s$ = 0.2401) |

## Requirements clustering

After mapping the software requirements into word embeddings, MAPP-Reviews uses the k-means algorithm (*MacQueen, 1967*) to obtain a clustering model of semantically similar software requirements.

Formally, let $R = \{r_1, r_2, \ldots, r_n\}$ a set of extracted software requirements, where each requirement $r$ is a $m$-dimensional real vector from an word embedding space. The k-means clustering aims to partition the $n$ requirements into $k$ ($2 \le k \le n$) clusters $C = \{C_1, C_2, \ldots, C_k\}$, thereby minimizing the within-cluster sum of squares as defined in Eq. (2), where $\mu_i$ is the mean vector of all requirements in $C_i$.

$$\sum_{C_i \in C} \sum_{r \in C_i} \|r - \mu_i\|^2 \qquad (2)$$

We observe that not all software requirements cluster represents a functional requirement in practice. Then, we evaluated the clustering model using a statistical measure called silhouette (*Rousseeuw, 1987*) to discard clusters with many different terms and irrelevant requirements. The silhouette value of a data instance is a measure of how similar a software requirement is to its own cluster compared to other clusters. The silhouette measure ranges from −1 to +1, where values close to +1 indicate that the requirement is well allocated to its own cluster (*Vendramin, Campello & Hruschka, 2010*). Finally, we use the requirements with higher silhouette values to support the cluster labeling, *i.e.*, to determine the software requirement's cluster name. For example, Table 3 shows the software requirement cluster "Payment" and some tokens allocated in the cluster with their respective silhouette values.

To calculate the silhouette measure, let $r_i \in C_i$ a requirement $r_i$ in the cluster $C_i$. Equation (3) compute the mean distance between $r_i$ and all other software requirements in the same cluster, where $d(r_i, r_j)$ is the distance between requirements $r_i$ and $r_j$ in the cluster $C_i$. In the equation, the expression $\frac{1}{|C_i|-1}$ means the distance $d(r_i, r_i)$ is not added to the sum. A smaller value of the silhouette measure $a(i)$ indicates that the requirement $i$ is far from neighboring clusters and better assigned to its cluster.

$$a(r_i) = \frac{1}{|C_i| - 1} \sum_{r_j \in C_i, r_i \neq r_j} d(r_i, r_j) \qquad (3)$$

Analogously, the mean distance from requirement $r_i$ to another cluster $C_k$ is the mean distance from $r_i$ to all requirements in $C_k$, where $C_k \neq C_i$. For each requirement $r_i \in C_i$, Eq. (4) defines the minimum mean distance of $r_i$ for all requirements in any other cluster,

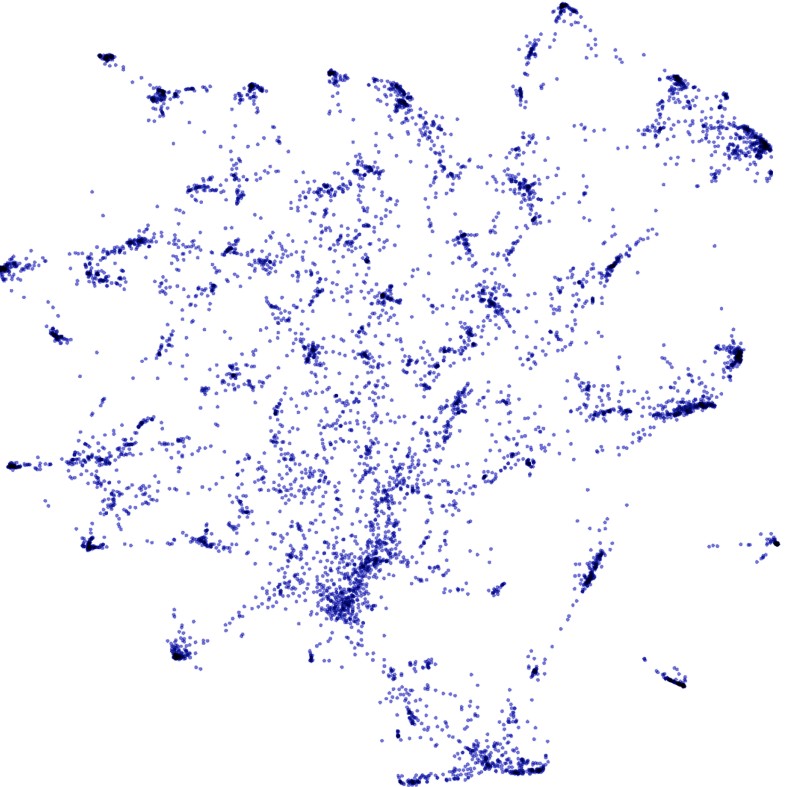

**Figure 4 Two-dimensional projection of clustered software requirements from approximately 85,000 food delivery app reviews.**

of which $r_i$ is not a member. The cluster with this minimum mean distance is the neighbor cluster of $r_i$. So this is the next best-assigned cluster for the $r_i$ requirement. The silhouette (value) of the software requirement $r_i$ is defined by Eq. (5).

$$b(r_i) = \min_{k \neq i} \frac{1}{|C_k|} \sum_{r_j \in C_k} d(r_i, r_j) \tag{4}$$

$$s(r_i) = \frac{b(r_i) - a(r_i)}{\max\{a(r_i), b(r_i)\}}, \text{ if } |C_i| > 1 \tag{5}$$

At this point in the MAPP-Reviews method, we have software requirements pre-processed and represented through contextual word embeddings, as well as an organization of software requirements into $k$ clusters. In addition, each cluster has a representative text (cluster label) obtained according to the requirements with higher silhouette values.

Figure 4 shows a two-dimensional projection of clustered software requirements from approximately 85,000 food delivery app reviews, which were used in the experimental evaluation of this work. High-density regions represent clusters of similar requirements that must be mapped to the same software requirement during the analysis of temporal dynamics. In the next section, techniques for generating the time series from software requirements clusters are presented, as well as the predictive models to infer future trends.

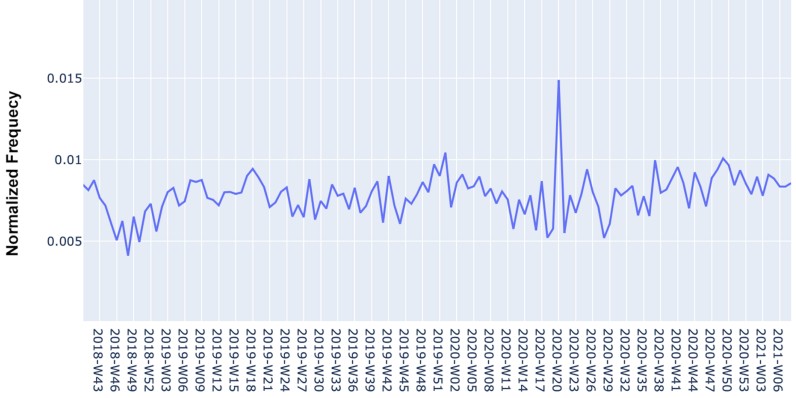

**Figure 5 Time series with the normalized frequency of "Arriving time" requirement from Zomato App in negative reviews.**

## Time series generation

Time series can be described as an ordered sequence of observations (*Chatfield & Xing, 2019*). A time series of size $s$ is defined as $X = (x_1, x_2,...,x_s)$ in which $x_t \in \mathbb{R}$ represents an observation at time $t$.

MAPP-Reviews generates time series for each software requirements cluster, where the observations represent how many times each requirement occurred in a period. Consequently, we know how many times a specific requirement was mentioned in the app reviews for each period. Each series models the temporal dynamics of a software requirement, *i.e.*, the temporal evolution considering occurrences in negative reviews.

Some software requirements are naturally more frequent than others, as well as the tokens used to describe these requirements. For the time series analysis to be compared uniformly, we generate a normalized series for each requirement. Each observation in the time series is normalized according to Eq. (6),

$$x_{normalized} = \frac{x}{z_p} \tag{6}$$

where $x_{normalized}$ is the result of the normalization, where $x$ is the frequency of cluster (time series observation) $C$ in the period $p$, $z_p$ is the total frequency of the period.

Figure 5 shows an example of one of the generated time series for a software requirement. The time dynamics represented in the time series indicate the behavior of the software requirement concerning negative reviews. Note that in some periods there are large increases in the mention of the requirement, thereby indicating that users have negatively evaluated the app for that requirement. Predicting the occurrence of these periods for software maintenance, aiming to minimize the number of future negative reviews is the objective of the MAPP-Reviews predictive model discussed in the next section.

## Predictive models

Predictive models for time series are very useful to support an organization in its planning and decision-making. Such models explore past observations to estimate observations in

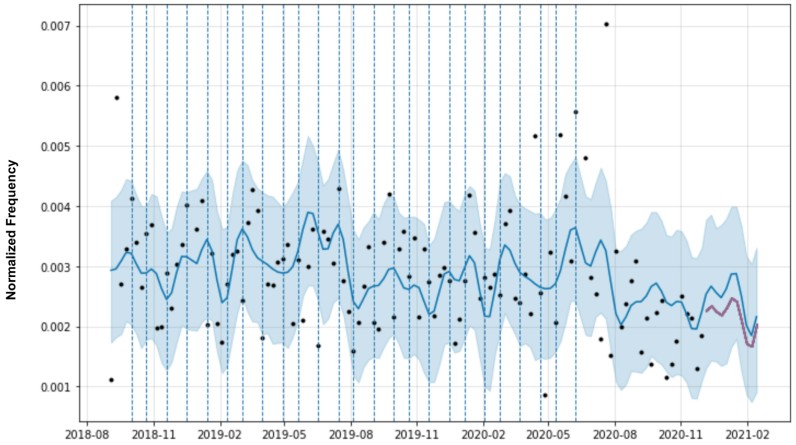

**Figure 6 Prophet forecasting with automatic change points of a requirement.**

future horizons, given a confidence interval. In our MAPP-Reviews method, we aim to detect the negative reviews of a software requirement that are starting to happen and make a forecast to see if they will become serious in the subsequent periods, *i.e.*, high frequency in negative reviews. The general idea is to use *p* points from the time series to estimate the next *p* + *h* points, where *h* is the prediction horizon.

MAPP-Reviews uses the Prophet Forecasting Model (*Taylor & Letham, 2018*). Prophet is a model from Facebook researchers for forecasting time series data considering non-linear trends at different time intervals, such as yearly, weekly, and daily seasonality. We chose the Prophet model for the MAPP-Reviews method due to the ability to incorporate domain knowledge into the predictive model. The Prophet model consists of three main components, as defined in Eq. (7),

$$y(t) = g(t) + s(t) + h(t) + t\varepsilon \qquad (7)$$

where $g(t)$ represents the trend, $s(t)$ represents the time series seasonality, $h(t)$ represents significant events that impacts time series observations, and the error term $t$ represents noisy data.

During model training, a time series can be divided into training and testing. The terms $g(t)$, $s(t)$ and $h(t)$ can be automatically inferred by classical statistical methods in the area of time series analysis, such as the Generalized Additive Model (GAM) (*Hastie & Tibshirani, 1987*) used in Prophet. In the training step, the terms are adjusted to find an additive model that best fits the known observations in the training time series. Next, we evaluated the model in new data, *i.e.*, the testing time series.

In the case of the temporal dynamics of the software requirements, domain knowledge is represented by specific points (*e.g.* changepoints) in the time series that indicate potential growth of the requirement in negative reviews. Figure 6 shows the forecasting for a software requirement. Original observations are the black dots and the blue line represents the forecast model. The light blue area is the confidence interval of the predictions. The vertical dashed lines are the time series changepoints.

Changepoints play an important role in forecasting models, as they represent abrupt changes in the trend. Changepoints can be estimated automatically during model training, but domain knowledge, such as the date of app releases, marketing campaigns, and server failures, are changepoints that can be added manually by software engineers. Therefore, the changepoints could be specified by the analyst using known dates of product launches and other growth-altering events or may be automatically selected given a set of candidates. In MAPP-Reviews, we have two possible options for selecting changepoints in the predictive model. The first option is automatic changepoint selection, where the Prophet specifies 25 potential changepoints which are uniformly placed in the first 80% of the time series. The second option is the manual specification which has a set of dates provided by a domain analyst. In this case, the changepoints could be entirely limited to a small set of dates. If no known dates are provided, by default we use the most recent observations which have a value greater than the average of the observations, *i.e.*, we want to emphasize the highest peaks of the time series, as they indicate critical periods of negative revisions from the past.

In the experimental evaluation, we show the MAPP-Reviews ability to predict perceptually important points in the software requirements time series, allowing the identification of initial trends in defective requirements to support preventive strategies in software maintenance.

Table 4 shows an emerging issue being predicted 6 weeks in advance in the period from October 2020 to January 2021. The table presents a timeline represented by the horizon ($h$) in weeks, with the volume of negative raw reviews (*Vol.*). An example of a negative review is shown for each week until reaching the critical week (peak), with $h = 16$. The table row with $h = 10$ highlighted in bold shows when MAPP-Reviews identified the uptrend. In this case, we show the MAPP-Reviews alert for the "*Time of arrival*" requirement of the Uber Eats app. In particular, the emerging issue identified in the negative reviews is the low accuracy of the estimated delivery time in the app. The text of the user review samples has been entered in its entirety, without any pre-treatment. A graphical representation of this prediction is shown in Fig. 7.

## RESULTS

The proposed approach is validated through an experimental evaluation with popular food delivery apps. These apps represent a dynamic and complex environment consisting of restaurants, food consumers, and drivers operating in highly competitive conditions (*Williams et al., 2020*). In addition, this environment means a real scenario of commercial limitations, technological restrictions, and different user experience contexts, which makes detecting emerging issues early an essential task. For this experimental evaluation, we used a dataset with 86,610 reviews of three food delivery apps: Uber Eats, Foodpanda, and Zomato. The dataset was obtained in the first stage (App Reviews) of MAPP-Reviews and is available at https://github.com/vitormesaque/mapp-reviews. The choice of these apps was based on their popularity and the number of reviews available. The reviews are from September 2018 to January 2021.

**Table 4 Example of emerging issue prediction alert for the "Time of arrival" requirement of the Uber Eats app reviews triggered by MAPP-reviews.**

| H | Vol. | Token | Review |
|---|---|---|---|
| 1 | 768 | Delivery time | Listed delivery times are inaccurate majority of the time. |
| 2 | 849 | Time frame | This app consistently gives incorrect, shorter delivery time frame to get you to order, but the deliveries are always late. The algorithm to predict the delivery time should be fixed so that you'll stop lying to your customers. |
| 3 | 896 | Arrival time | Ordered food and they told me it was coming. The wait time was supposed to be 45 min. They kept pushing back the arrival time, and we waited an hour and 45 min for food, only to have them CANCEL the order and tell us it wasn't coming. If an order is unable to be placed you need to tell customers BEFORE they've waited almost 2 h for their food. |
| 4 | 1247 | Delivery time | The app was easy to navigate but the estimated delivery time kept changing and it took almost 2 h to receive food and I live less than 4 blocks away pure ridiculousness if I would of know that I would of just walked there and got it. |
| 5 | 1056 | Estimated time | Everyone cancels and it ends up taking twice the estimated time to get the food delivered. You don't get updated on delays unless you actively monitor. Uber has failed at food delivery. |
| 6 | 997 | More time | Uber Eats lies. Several occasions showed delays because "the restaurant requested more time" but really it was Uber Eats unable to find a driver. I called the restaurants and they said the food has been ready for over an hour! |
| 7 | 939 | Delivery time | Your app is unintuitive. Delivery times are wildly inaccurate and orders are cancelled with no explanation, information or help. |
| 8 | 854 | Estimated time | This service is terrible. Delivery people never arrive during the estimated time. |
| 9 | 994 | Time | Delivery times increase significantly once your order is accepted. 25–45 min went up to almost 2 h! Not easy to cancel. Also one restaurant that looked available said I was too far away after I had filled my basket. Other than that the app is easy to use. |
| 10 | 1257 | Time estimate | Use door dash or post mates, Uber eats has definitely gone down in quality. Extremely inaccurate time estimates and they ignore your support requests until its to late to cancel an order and get a refund. |
| 11 | 1443 | Delivery time | Delivery times are constantly updated, what was estimated at 25-35 min takes more than 2 h. I understand it's just an estimate, but 4X that is ridiculous. |
| 12 | 1478 | Delivery time | Inaccurate delivery time |
| 13 | 1376 | Estimated time | Used to use this app a lot. Ever since they made it so you have to pay for your delivery to come on time the app is useless. You will be stuck waiting for food for an hour most of the time. The estimated time of arrival is never accurate. Have had my food brought to wrong addresses or not brought at all. I will just take the extra time out of my day to pick up the food myself rather than use this app. |
| 14 | 1446 | Estimated time | Terrible, the estimated time of arrival is never accurate and has regularly been up to 45 MINUTES LATE with no refund. Doordash is infinitely better, install that instead, it also has more restaurants |
| 15 | 1354 | Estimated time | App is good but this needs to be more reliable on its service. the estimated arrival time needs to be matched or there should be a option to cancel the order if they could not deliver on estimated time. Continuously changing the estimated delivery time after the initial order confirmation is inappropriate. |
| 16 | 1627 | Estimated time | I use this app a lot and recently my order are always late at least double the time in originally quoted. Every time my food is cold. Maybe the estimated time should be adjusted to reflect what the actual time may be. |

After the software requirements extraction and clustering stage (with $k = 300$ clusters), the six most popular (frequent) requirements clusters were considered for time series prediction. The following software requirements clusters were selected: "Ordering", "Go pick up", "Delivery", "Arriving time", "Advertising", and "Payment". The requirements clusters are shown in Table 5 with the associated words ordered by silhouette.

In the MAPP-Reviews prediction stage, we evaluated two scenarios using Prophet. The first scenario is the baseline, where we use the automatic parameters fitting of the Prophet. By default, Prophet will automatically detect the changepoints. In the second scenario, we

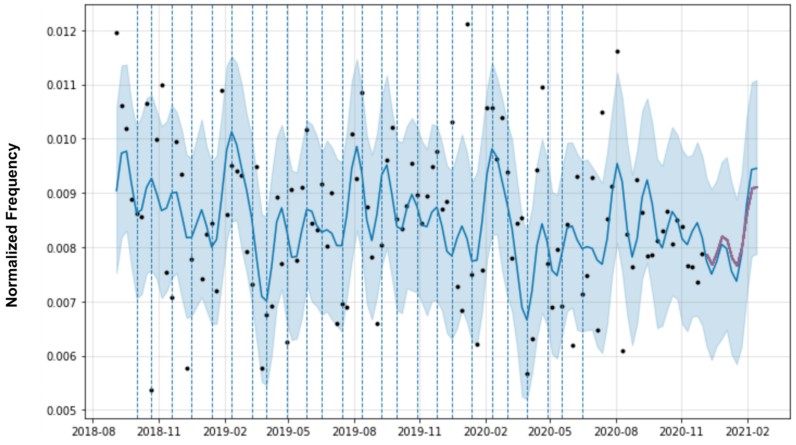

**Figure 7 Forecasting for software requirement cluster (Arriving time) from Uber Eats app reviews.**

**Table 5 Software requirements clusters for food delivery apps used in the experimental evaluation.** Tokens well allocated in each cluster (silhoutte measure) were selected to support the cluster labeling.

| Cluster label | Tokens with Silhouette values (s) |
|---|---|
| Ordering | "ordering" ($s = 0.1337$), "order's" ($s = 0.1250$), "order from" ($s = 0.1243$), "order will" ($s = 0.1221$), "order" ($s = 0.1116$), "the order" ($s = 0.1111$) |
| Go pick up | "go pick up" ($s = 0.1382$), "pick up the" ($s = 0.1289$)", "pick up at" ($s = 0.1261$), "to take" ($s = 0.1176$), "go get" ($s = 0.1159$) |
| Delivery | "delivering parcels" ($s = 0.1705$), "delivery options" ($s = 0.1590$), "waive delivery" ($s = 0.1566$), "delivery charges" ($s = 0.1501$), "accept delivery" ($s = 0.1492$) |
| Arriving time | "arrival time" ($s = 0.3303$), "waisting time" ($s = 0.3046$), "arriving time" ($s = 0.3042$), "estimate time" ($s = 0.2877$), "delivery time" ($s = 0.2743$) |
| Advertising | "anoyning ads" ($s = 0.3464$), "pop-up ads" ($s = 0.3440$), "ads pop up" ($s = 0.3388$), "commercials advertise" ($s = 0.3272$), "advertising" ($s = 0.3241$) |
| Payment | "payment getting" ($s = 0.2618$), "payment get" ($s = 0.2547$), "getting payment" ($s = 0.2530$), "take payment" ($s = 0.2504$), "payment taking" ($s = 0.2471$), "payment" ($s = 0.2401$) |

specify the potential changepoints, thereby providing domain knowledge for software requirements rather than automatic changepoint detection. Therefore, the changepoint parameters are used when we provide the dates of the changepoints instead of the Prophet determining them. In this case, we use the most recent observations that have a value greater than the average of observations, *i.e.*, critical periods with high frequencies of negative reviews in the past.

We used the MAPE (Mean Absolute Percentage Error) metric to evaluate the forecasting performance (*Makridakis, 1993*), as defined in Eq. (8),

$$MAPE = \frac{1}{h}\sum_{t=1}^{h} \frac{|real_t - pred_t|}{real_t} \qquad (8)$$

where $real_t$ is the real value and $pred_t$ is the predicted value by the method, and $h$ is the number of forecast observations in the estimation period (prediction horizon). In practical terms, MAPE is a measure of the percentage error that, in a simulation, indicates how close

**Table 6 Comparison of MAPE in general.**

| h | MAPE (Mean ± SD) | |
| --- | --- | --- |
| | (1) Automatic changepoint | (2) Specifying the changepoints |
| 1 | 13.82 ± 16.42 | 15.47 ± 14.42 |
| 2 | 15.58 ± 19.09 | 16.94 ± 17.20 |
| 3 | 16.26 ± 20.18 | 17.60 ± 18.71 |
| 4 | 16.09 ± 19.24 | 17.47 ± 18.37 |

**Table 7 MAPE analysis (at the peaks of the time series) of each scenario considering the software requirements.**

| h | MAPE (Mean ± SD) | |
| --- | --- | --- |
| | (1) Automatic changepoint | (2) Specifying the changepoints |
| 1 | 10.65 ± 8.41 | 10.30 ± 8.06 |
| 2 | 11.61 ± 8.80 | 11.00 ± 8.71 |
| 3 | 11.81 ± 8.86 | 11.42 ± 8.52 |
| 4 | 11.49 ± 8.71 | 11.19 ± 8.34 |

the prediction was made to the known values of the time series. We consider a prediction horizon ($h$) ranging from 1 to 4, with weekly seasonality.

Table 6 summarizes the main experimental results. The first scenario (1) with the default parameters obtains superior results compared to the second scenario (2) for all forecast horizons. In general, automatic changepoints obtains 9.33% of model improvement, considering the average of MAPE values from all horizons ($h = 1$ to $h = 4$).

In particular, we are interested in the peaks of the series since our hypothesis is that the peaks represent potential problems in a given software requirement. Thus, Table 7 shows MAPE calculated only for time series peaks during forecasting. In this case, predictions with the custom changepoints locations (scenario 2) obtained better results than the automatic detection for all prediction horizons ($h = 1$ to $h = 4$), obtaining 3.82% of forecasting improvement. These results provide evidence that domain knowledge can improve the detection of potential software requirements to be analyzed for preventive maintenance.

In particular, analyzing the prediction horizon, the results show that the best predictions were obtained with $h = 1$ (1 week). In practical terms, this means the initial trend of a defective requirement can be identified 1 week in advance.

Finally, to exemplify MAPP-Reviews forecasting, Fig. 7 shows the training data (Arriving time software requirement) represented as black dots and the forecast as a blue line, with upper and lower bounds in a blue shaded area. At the end of the time series, the darkest line is the real values plotted over the predicted values in blue. The lines plotted vertically represent the changepoints.

For reproducibility purposes, we provide a GitHub repository at https://github.com/vitormesaque/mapp-reviews containing the source code and details of each stage of the method, as well as the raw data and all the results obtained.

## DISCUSSION

Timely and effective detection of software requirements issues is crucial for app developers. The results show that MAPP-Reviews can detect significant points in the time series that provide information about the future behavior of a software requirement, allowing software engineers to anticipate the identification of emerging issues that may affect app evaluation. An issue related to a software requirement reported in user reviews is defined as an emerging issue when there is an upward trend for that requirement in negative reviews. Our method trains predictive models to identify requirements with higher negative evaluation trends, but a negative review will inevitably impact the rating. However, our objective is to mitigate this negative impact.

The prediction horizon ($h$) is an essential factor in detecting emerging issues to mitigate negative impacts. Software engineers and the entire development team need to know as soon as possible about software problems to anticipate them. In this context would not be feasible to predict the following months as it is tough to find a correlation between what happens today and what will happen in the next few months about bug reports. Therefore, MAPP-Reviews forecasts at the week level. This strategy allows us to identify the issues that are starting to happen and predict whether they will worsen in the coming weeks. Even at the week level, the best forecast should be with the shortest forecast horizon, *i.e.*, 1 week ($h = 1$). A longer horizon, *i.e.*, three ($h = 3$) or 4 weeks ($h = 4$), could be too late to prevent an issue from becoming severe and having more impact on the overall app rating. The experimental evaluation shows that our method obtains the best predictions with the shortest horizon ($h = 1$). In practical terms, this means that MAPP-Reviews identifies the initial trend of a defective requirement a week in advance. In addition, we can note that a prediction error rate (MAPE) of up to 20% is acceptable. For example, consider that the prediction is 1,000 negative reviews for a specific requirement at a given point, but the model predicts 800 negative reviews. Even with 20% of MAPE, we can identify a significant increase in negative reviews for a requirement and trigger alerts for preventive software maintenance, *i.e.*, when MAPP-Reviews predicts an uptrend, the software development team should receive an alert. In the time series forecast shown in Fig. 7, we observe that the model would be able to predict the peaks of negative reviews for the software requirement 1 week in advance.

The forecast presented in Fig. 7 shows that the model was able to predict the peak of negative reviews for the "Arriving time" requirement. An emerging issue detection system based only on the frequency of a topic could trigger many false detections, *i.e.*, it would not detect defective functionality but issues related to the quality of services offered. Analyzing user reviews, we found that some complaints are about service issues rather than defective requirements. For example, the user may complain about the delay in the delivery service and negatively rate the app, but in reality, they are complaining about the restaurant, *i.e.*, a problem with the establishment service. We've seen that this pattern of user complaints is repeated across other app domains, not just the food delivery service. In delivery food apps, these complaints about service are constant, uniform, and distributed among all restaurants available in the app. In Table 4, it is clear that the

emerging issue refers to the deficient implementation of the estimated delivery time prediction functionality. Our results show that when there is a problem in the app related to a defective software requirement, there are increasing complaints associated with negative reviews regarding that requirement.

An essential feature in MAPP-Reviews is changepoints. Assume that a time series represents the evolution of a software requirement over time, observing negative reviews for this requirement. Also, consider that time series frequently have abrupt changes in their trajectories. Given this, the changepoints describe abrupt changes in the time series trend, *i.e.*, means a specific date that indicates a trend change. Therefore, specifying custom changepoints becomes significantly important for the predictive model because the uptrend in time series can also be associated with domain knowledge factors. By default, our model will automatically detect these changepoints. However, we have found that specifying custom changepoints improves prediction significantly in critical situations for the emerging issue detection problem. In general, the automatic detection of changepoints had better MAPE results in most evaluations. However, the custom changepoints obtained the best predictions at the time series peaks for all horizons ($h = 1$ to $h = 4$) of experiment simulations. Our experiment suggests a greater interest in identifying potential defective requirements trends in the time series peaks. As a result, we conclude that specifying custom changepoints in the predictive model is the best strategy to identify potential emerging issues.

Furthermore, the results indicate the potential impact of incorporating changepoints into the predictive model using the information of app developers, *i.e.*, defining specific points over time with a meaningful influence on app evaluation. In addition, software engineers can provide sensitive company data and domain knowledge to explore and improve the predictive model potentially. For this purpose, we depend on sensitive company data related to the software development and management process, *e.g.*, release planning, server failures, and marketing campaigns. In particular, we can investigate the relationship between the release dates of app updates and the textual content of the update publication with the upward trend in negative evaluations of a software requirement. In a real-world scenario in the industry, software engineers using MAPP-Reviews will provide domain-specific information.

We show that MAPP-Reviews provides software engineers with tools to perform software maintenance activities, particularly preventive maintenance, by automatically monitoring the temporal dynamics of software requirements.

The results of our research show there are new promising prospects for the future, and new possibilities for innovation research in this area emerge with our results so far. We intend to explore further our method to deeply determine the input variables that most contribute to the output behavior and the non-influential inputs or to determine some interaction effects within the model. In addition, sensitivity analysis can help us reduce the uncertainties found more effectively and calibrate the model.

## Limitations

Despite the significant results obtained, we can still improve the predictive model. In the scope of our experimental evaluation, we only investigate the incorporation of software

domain-specific information through trend changepoints. Company-sensitive information and the development team's domain knowledge were not considered in the predictive model because we don't have access to this information. Therefore, we intend to evaluate our proposed method in the industry and explore more specifics of the domain knowledge to improve the predictive model.

Another issue that is important to highlight is the sentiment analysis in app reviews. We assume that it is possible to improve the classification of negative reviews by incorporating sentiment analysis techniques. We can incorporate a polarity classification stage (positive, negative, and neutral) of the extracted requirement, allowing a software requirements-based sentiment analysis. In the current state of our research, we only consider negative reviews with low ratings and associate them with the software requirements mentioned in the review.

Finally, to use MAPP-Reviews in a real scenario, there must be already a sufficient amount of reviews distributed over time, *i.e.*, a minimum number of time-series observations available for the predictive model to work properly. Therefore, in practical terms, our method is more suitable when large volumes of app reviews are available to be analyzed.

## CONCLUSIONS

Opinion mining for app reviews can provide useful user feedback to support software engineering activities. We introduced the temporal dynamics of requirements analysis to predict initial trends on defective requirements from users' opinions before negatively impacting the overall app's evaluation. We presented the MAPP-Reviews (Monitoring App Reviews) approach to (1) extract and cluster software requirements, (2) generate time series with the time dynamics of requirements, (3) identify requirements with higher trends of negative evaluation.

The experimental results show that our method is able to find significant points in the time series that provide information about the future behavior of a requirement through app reviews, thereby allowing software engineers to anticipate the identification of requirements that may affect the app's evaluation. In addition, we show that it's beneficial to incorporate changepoints into the predictive model by using domain knowledge, *i.e.*, defining points over time with significant impacts on the app's evaluation.

We compared the MAPP-Reviews in two scenarios: first using automatic changepoint detection and second specifying the changepoint locations. In particular, the automatic detection of points of change had better MAPE results in most evaluations. On the other hand, the best predictions at the time series peaks (where there is a greater interest in identifying potential defective requirements trends) were obtained by specifying changepoints.

Future work directions involve evaluating MAPP-Reviews in other scenarios to incorporate and compare several other types of domain knowledge into the predictive model, such as new app releases, marketing campaigns, server failures, competing apps, among other information that may impact the evaluation of apps. Another direction for

future work is to implement a dashboard tool for monitoring app reviews, thus allowing the dispatching of alerts and reports.

### Funding
This study was supported by the Brazilian National Council for Scientific and Technological Development (CNPq) [process number 426663/2018-7], the Federal University of Mato Grosso do Sul (UFMS), the São Paulo Research Foundation (FAPESP) [process number 2019/25010-5 and 2019/07665-4], BIRDIE.AI (Project CEIA/UFG-PEIA-2105.0011), and the Brazilian Company of Research and Industrial Innovation (EMBRAPII). The funders had no role in study design, data collection and analysis, decision to publish, or preparation of the manuscript.

### Grant Disclosures
The following grant information was disclosed by the authors:
Brazilian National Council for Scientific and Technological Development (CNPq): 426663/2018-7.
Federal University of Mato Grosso do Sul (UFMS), São Paulo Research Foundation (FAPESP): 2019/25010-5 and 2019/07665-4.
BIRDIE.AI: CEIA/UFG-PEIA-2105.0011.
Brazilian Company of Research and Industrial Innovation (EMBRAPII).

### Competing Interests
The authors declare that they have no competing interests.

### Author Contributions
- Vitor Mesaque Alves de Lima conceived and designed the experiments, performed the experiments, analyzed the data, performed the computation work, prepared figures and/or tables, authored or reviewed drafts of the paper, and approved the final draft.
- Adailton Ferreira de Araújo conceived and designed the experiments, analyzed the data, performed the computation work, authored or reviewed drafts of the paper, and approved the final draft.
- Ricardo Marcondes Marcacini performed the computation work, authored or reviewed drafts of the paper, and approved the final draft.

### Data Availability
The data and codes are available at GitHub: https://github.com/vitormesaque/mapp-reviews.

### Supplemental Information
Supplemental information for this article can be found online at http://dx.doi.org/10.7717/peerj-cs.874#supplemental-information.

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
