# Peer review of "Temporal dynamics of requirements engineering from mobile app reviews"

_PeerJ Computer Science, doi:10.7717/peerj-cs.874_

## Round 0.1 · original submission · Major Revisions

I have received detailed reports on the paper. Please provide responses to the comments. Note that I do not expect you to cite any reference recommended by reviewers unless it is relevant. Not including recommended references will not affect my editorial decision.

Reviewer 1 ·

Basic reporting

Line 73: reviews. First, we collect, pre-process and extract software requirements from large review datasets. The authors might need to the name of the apps and explain the process in details in section 3.

Line 74: Then, the software requirements associated with negative reviews are organized into groups according to their content similarity. The authors might need to explain how

Figure 1. : It is very high level. Lack of analytical details need to show the steps in more explicit way.

Line 201: The reviews are from 8 apps of different categories. What are the name of these apps and their categories?

Line 318-9: It would be better to move it up. After you introduce the model.

Line 321: Does it mean that you consider the negative review as the change-point?. This part is lack of a clear explanation. Moreover, it needs more supporting example (e.g., why did the author consider the negative review as the change-point)

Experimental design

Line 76 and 217: software requirement from negative reviews: The authors need to explain how they classify the negative reviews in more details (not only by considering the low rate as negative), as well as The authors, might need to read the reviews to check if it can be considered as negative or positive reviews, even if the rate is low/high (1/5). Example, the reviewers might rate the app 3 out of 5 and it might be considered as negative review?

Line 219: Thus, we use RE-BERT to extract …..…... It is not clear how this step is preformed? How did the authors classify them into complaints, bad usage experience, or malfunction

Line 193: Does the training data contain negative review? How would the model be able to predict them ? The authors need to explain these points.

Line 233: It is not clear how BERT would help your model? why do you need it at the first place?

Line 234: It is not clear how this objective is related to the proposed model.

Validity of the findings

In general: Lack of a comprehensive experiment. The model should be evaluated with some of the existing solutions to see how its performance.

Line 355: we used a dataset with 86,610 reviews of: Did they collect the datasets or is it publicly available? if it is publicly available where is the reference and why did they choose it? What is the dimensionality of the dataset? What is the name of these apps.

Line 358: clustering stage (with k = 300 clusters) why k=300

Reviewer 2 ·

Basic reporting

The paper does not follow the standard sections for the journal. Furthermore, Acknowledgments acknowledge funders as additional contradiction to the guidelines. The authors do not share any raw data on their evaluation. Neither the app review nor the results are shared. Therefore, none of their results can be judged.

Experimental design

no comment

Validity of the findings

The data used for the evaluation was not made available. Therefore the validity cannot be judged. Explaining the data set as well as sharing it online would improve the paper a lot. Furthermore, the authors could also publish the results on the data when having applied the MAPP method.

Additional comments

The paper covers an interesting and promising topic. It is written in a clear and well understandable language. The two mentioned problems can easily be addressed with a revision of the paper.
As minor addition I would also suggest to extend the motivation with references that support the goal of continuous evaluation of user feedback side by side to the development process like T Palomba et al. [1] and Hassan et al. [2] as well as Scherr [3] et al. do.
1. Palomba, F., Linares-Vásquez, M., Bavota, G., Oliveto, R., Di Penta, , Poshyvanyk, D., De Lucia, A.: Crowdsourcing User Reviews to Support the Evolution of Mobile Apps. Journal of Systems and Software(March 2018), 143-162 DOI:10.1016/j.jss.2017.11.043 (137)
2. Hassan, S., Tantithamthavorn, C., Bezemer, C.-P., Hassan, A.: Studying the dialogue between users and developers of free apps in the Google Play Store. Empirical Software Engineering 23(3), 1275–1312 https://doi.org/10.1007/s10664-017-9538-9 (2018)
3. Scherr, S., Hupp, S., Elberzhager, F.: Establishing Continuous App Improvement by Considering Heterogenous Data Sources. International Journal of Interactive Mobile Technologies (iJIM) 15(10), 66-86 (2021)

Reviewer 3 ·

Basic reporting

- Missing the threats to validity and/or limitation section.
- Missing the discussions and/or implications section.
- It still isn’t clear to me how this can be used concretely as this is not detecting emerging issues, but rather predicting the future frequency of a requirement in negative reviews. Could you elaborate more on why this prediction is needed as the requirements from negative reviews should have already impacted the app's rating? Additionally, as a developer looking at the results, when would I say that the problem/requirement is becoming major? (e.g., the slope of change/the change in the frequency is greater than some particular threshold?). These could be addressed in the discussions and-or implications section.

Experimental design

- Some methodological decisions need more details (e.g., I’m not quite sure how you decide the starting number of k? does selecting a different starting number k affect the result in any way? what's the rationale behind selecting six clusters in your experiment? Can a requirement belong to more than one cluster?)
- Were reviewed cleaned (e.g., based on length?).
- Your experiment should also include using app update dates as changepoints because prior research suggested that naturally, you will see a spike of user reviews shortly after app updates.

Validity of the findings

- For a journal paper, a further extensive evaluation is needed. I would argue that reviews from food ordering apps did not contain rich data regarding the problems/requirements with the app itself but contain a great number of information/complaints on the services.
- How applicable can this be used for apps in other categories? Your experimentation is done only on one type of apps and only popular ones (please discuss this in the threats to validity and/or limitation section).
- The experimental evaluation is quite lacking compared to other sections. You could revisit the research question you introduced in the introduction.
- Sensitivity analysis may be needed.

Additional comments

- Interesting research, relevant to the journal.
- It is well written, structured, and presented.
- Extensive literature review.
- The authors provided a code to examine.

---

## Round 0.2 · Minor Revisions

A minor revision is recommended. I hope to see your revision soon.

Reviewer 1 ·

Basic reporting

All good

Experimental design

All good

Validity of the findings

All good

Additional comments

All good

Reviewer 2 ·

Basic reporting

no comment

Experimental design

no comment

Validity of the findings

The findings are explained in a more solid way due to the introduction of a discussion section. Nevertheless half of this rather short section is spent on the limitations and not on discussion the findings. I would like to see a deeper discussions of the interesting findings.

Additional comments

The paper covers an interesting and promising topic. It is written in a clear and well understandable
language. The authors provided a significant improvement over the first version.

---

## Round 0.3 · accepted · Accept

I am satisfied with the revision and the responses. I recommend it for publication.